# The Impact of Mode of Birth on Childbirth-Related Post Traumatic Stress Symptoms beyond 6 Months Postpartum: An Integrative Review

**DOI:** 10.3390/ijerph19148830

**Published:** 2022-07-20

**Authors:** Nicole Ginter, Lea Takács, Martine J. M. Boon, Corine J. M. Verhoeven, Hannah G. Dahlen, Lilian L. Peters

**Affiliations:** 1Department of General Practice & Elderly Medicine, Section Midwifery Science, University Medical Center Groningen, University of Groningen, 9700 RB Groningen, The Netherlands; m.j.m.boon@student.rug.nl (M.J.M.B.); l.l.peters@umcg.nl (L.L.P.); 2Department of Midwifery Science, AVAG, Amsterdam Public Health Research Institute, Amsterdam UMC, Vrije Universiteit Amsterdam, 1081 BT Amsterdam, The Netherlands; c.verhoeven@amsterdamumc.nl; 3Department of Psychology, Faculty of Arts, Charles University, 116 42 Prague, Czech Republic; lea.takacs@ff.cuni.cz; 4Division of Midwifery, School of Health Sciences, University of Nottingham, Nottingham NG7 2HA, UK; 5Department of Obstetrics and Gynaecology, Maxima Medical Centre, 5504 DB Veldhoven, The Netherlands; 6School of Nursing and Midwifery, Western Sydney University, Penrith, NSW 2751, Australia; h.dahlen@westernsydney.edu.au

**Keywords:** post-traumatic stress disorder, mode of birth, childbirth, postpartum, mothers

## Abstract

(1) Background: A traumatic birth can lead to the development of childbirth-related posttraumatic stress symptoms or disorder (CB-PTS/D). Literature has identified the risk factors for developing CB-PTS/D within the first six months postpartum thoroughly. However, the impact of mode of birth on CB-PTS/D beyond 6 months postpartum is scarcely studied. (2) Methods: A systematic search of the literature was conducted in the databases PubMed, Embase and CINAHL and PRISMA guidelines were followed. Studies were included if they reported the impact of mode of birth on CB-PTS/D beyond 6 months postpartum. (3) Results: In total, 26 quantitative and 2 qualitative studies were included. In the quantitative studies the percentage of women with CB-PTS/D ranged from 0.7% to 42% (between six months and five years postpartum). Compared with vaginal birth, operative vaginal birth, and emergency caesarean section were associated with CB-PTS/D beyond 6 months postpartum. Qualitative studies revealed that some women were suffering from CB-PTS/D as long as 18 years after birth. (4) Conclusions: Long- term screening of women for PTSD in the postnatal period could be beneficial. More research is needed on models of care that help prevent CB-PTS/D, identifying women at risk and factors that maintain CB-PTS/D beyond 6 months postpartum.

## 1. Introduction

Childbirth is one of the most profound experiences in a woman’s life and can be positive and empowering, but also stressful or even traumatic [1]. Although post-traumatic stress disorder (PTSD) was originally studied in military veterans, during the last two decades, childbirth has too been recognized as a potentially traumatic event that can lead to PTSD symptoms or even PTSD diagnosis. Experiencing fear and powerlessness during childbirth or, most profoundly, fearing for one’s own life or one’s own child dying can trigger a post-traumatic stress (PTS) response which presents with symptoms such as persistent involuntary and intrusive memories, nightmares, avoidance of certain stimuli, difficulties concentrating and hypervigilance [2,3]. According to the criteria of the Diagnostic and Statistical Manual (DSM-V), PTSD is diagnosed when those symptoms persist for at least one month and interfere with daily functioning. Studies reported that 2% to 9% of new mothers suffer from childbirth-related post-traumatic stress disorder (CB-PTS/D) assessed within a range of 4 weeks to 9 months after childbirth [4,5,6,7].

Childbirth related post traumatic stress symptoms and disorder (CB-PTS/D) do not only affect maternal wellbeing but may also have negative impact on the child and other family members [8]. Severe fear of reliving the traumatic event can result in fear of future childbirth, postponing another pregnancy, or ultimately not having any more children, despite a wish to have them [9]. 

Furthermore, traumatic childbirth experience can lead to a woman’s avoidance of intimacy, which could eventually be detrimental to the relationship with her partner [9]. Being around the newborn can trigger reliving traumatic birth memories repeatedly, which may have an impact on maternal bonding and result in the mother being emotionally unavailable and unresponsive to her child’s needs [10]. This can all contribute to poor infant sleep, disrupted eating patterns and suboptimal cognitive and emotional development [11,12]. 

Previous studies have linked CB-PTS/D with unexpected health complications and life threatening situations during childbirth [13,14,15]. Emergency caesarean section (emergency CS) and operative vaginal birth have particularly been found to be associated with CB-PTS/D [14,15], although research in this area is still emerging and the results of the studies are conflicting [16,17]. Nevertheless, not being able to give birth vaginally and having an unplanned CS can be experienced as traumatic, leading to CB-PTS/D [3,18]. As the CS rate almost doubled (rose from 12% to 21% worldwide) between 2000 and 2015 [19], examining the association between mode of birth and CB-PTS/D is warranted.

It must however be pointed out that CB-PTS/D may also develop as a response to physiological, uncomplicated vaginal birth if the subjective experience of childbirth is traumatic, for example due to lack of respect and involvement in decision-making, unsupportive attitude of caregivers and lack of emotional support provided to the laboring woman [4,20,21,22]. Existing studies report a higher risk of CB-PTS/D in primiparous compared to multiparous women [23,24]; other factors that increase the risk for CB-PTS/D include a history of mental health problems, such as depression and anxiety [13,25,26,27], previous PTSD [28] and trauma, such as sexual abuse or domestic violence [26,29].

Previous literature reviews on CB-PTS/D focused on prevention [30], prevalence and risk factors [31,32,33], impact on the couple’s relationship [34], and effects on child development [11,12]. Although several reviews included mode of birth as a potential predictor of CB-PTS/D, to date there is no published review assessing the association between CB-PTS/D and the different mode of birth specifically. A systematic review examining the association between mode of birth and PSTD in the first six months postpartum is underway, [35] but literature summarizing the knowledge regarding the impact of mode of birth on CB-PTS/D beyond this period is lacking. As several studies show a significant number of women suffering from CB-PTS/D [36,37,38] beyond the first six postpartum months, investigating the impact of birth mode on CB-PTS/D in the long term is warranted. 

It was therefore the aim of this integrative review to examine the impact of different modes of birth (e.g., vaginal birth, CS) on CB-PTS/D beyond six months postpartum. 

## 2. Materials and Methods

### 2.1. Type of Review and Search Strategy

To answer our research question, we conducted an integrative review according to the PRISMA guidelines [39]. We searched the following databases to identify articles potentially eligible for this review: PubMed, Embase, and CINAHL. The search included the following keywords: mode of birth (e.g., caesarean section, operative vaginal birth), postpartum period (e.g., postpartum), and PTSD (e.g., post-traumatic stress disorder/symptoms). 

The detailed search strategy is presented in the Appendix B. Apart from the studies identified through the systematic search of the databases, we also screened the references listed in the selected studies and added articles identified by a manual search. The search period for publications was limited to the last 30 years (from 1 January 1990 until 17 May 2022). This integrative review was registered in PROSPERO (CRD42020212745).

### 2.2. Inclusion and Exclusion Criteria

Both quantitative and qualitative studies were eligible for this review. We included English written published articles and did not include results of unpublished data or materials. Studies that recruited women during pregnancy or after childbirth were eligible. We excluded studies reporting results exclusively for women younger than 18 and/or for those with less favorable perinatal outcomes (such as postpartum hemorrhage and premature or low-birth weight newborns), as those conditions are associated with CB-PTS/D [40,41].

Quantitative studies were included if they reported CB-PTS/D measurement scores differentiated by mode of birth and correlations and/or (crude/adjusted) associations between CB-PTS/D and mode of birth. Studies reporting data on the following modes of birth were included: vaginal birth, operative vaginal birth, caesarean section (CS), emergency CS or elective CS.

We included studies assessing CB-PTS/D on the basis of interviews or questionnaires designed in accordance with DSM-IV or DSM-V criteria [42,43], that were either measures of PTSD related to childbirth specifically or measures of PTSD generally related to any traumatic event, with additional questions identifying childbirth as the traumatic event. 

Only studies reporting on CB-PTS/D at 6 months postpartum or later, or with a mean time point of 6 months postpartum or more, were included.

### 2.3. PTSD Symptomatology Criteria

According to the DSM-IV, the PTSD diagnosis is based on the following six criteria (A-F), out of which three are the symptom clusters (B-D). Criterion A identifies the traumatic event. Criterion B describes the re-experiencing/intrusion symptom cluster (e.g., nightmares or intrusive thoughts of the event), where at least one out of five symptoms need to be present to meet the criteria for PTSD diagnosis. Criterion C consists of seven avoidance symptoms (e.g., avoidance of thoughts, places or people associated with the trauma, or inability to recall aspects of the trauma) of which three need to be present. Criterion D describes five hyperarousal symptoms (e.g., difficulties concentrating and hypervigilance) of which two need to be present. Criterion E is fulfilled when the symptoms listed under B, C and D are present for at least one month, and the symptoms need to lead to clinically significant distress or impairment in functioning to fulfill criterion F [42]. Criteria A–F must be present to meet the diagnostic criteria for PTSD diagnosis. 

The criteria for PTSD diagnosis have been modified in the DSM-V published in 2013 [44]. 

In DSM-IV, criterion A, which identifies the traumatic event, consists of two parts: experiencing an event that involved an actual or threatened death or serious injury and the personal response to the event with fear, helplessness or horror. The personal response to the event was removed in DSM-V [43,45]. Additionally, a symptom cluster referring to negative alterations in cognition and mood (e.g., feelings of detachment, inability to experience positive emotions) was added in DSM-V [42,43]. A further description of the DSM-IV and DSM-V criteria and their differences is provided by Pai et al., and in the APA factsheet [43,45].

There are several validated questionnaires and structured interviews that can be used to assess PTSD in accordance with DSM-IV or DSM-V criteria. Although most of the existing questionnaires designed to measure PTSD are related to any type of trauma, there are few questionnaires developed specifically to measure CB-PTS/D [44]. 

### 2.4. Selection of Studies

References from the studies identified as potentially eligible for this review were exported to the bibliographic management software Endnote, where a removal of the duplicates was carried out. Subsequently, they were exported to Rayyan, a reference management app for data collection and initial screening of titles and abstracts [46]. Rayyan was used for overall management and storage of the studies including repeated deduplication and removal of non-English articles. A two-step selection process was adopted: in the first step, two researchers (NG, MB) independently screened the titles and abstracts of the articles for their eligibility. The second step consisted of evaluation of full texts of the studies regarding the inclusion and exclusion criteria. Any discrepancy between the researchers were solved by consensus or, if necessary, a third researcher (LT) was consulted to resolve uncertainties about eligibility. 

### 2.5. Quality Assessment

The quality of the studies was assessed by two researchers (NG, MB) independently using the Critical Appraisal Skills Programme (CASP) for quantitative and qualitative studies [47]. 

The CASP-scoring system developed by Barnett et al. (2012) was used to assess the quality of cohort and cross-sectional studies [48]. The CASP checklist assesses internal validity by appraising potential measurement biases and confounding, and external validity by appraising a potential selection bias and is therefore suitable for both cohort and cross-sectional studies [48,49] (Appendix A). 

The CASP-checklist comprised 12 items which were used to assess the overall quality and assessing the following criteria (1) selection bias; (2) exposure measurement bias (mode of birth); (3) outcome measurement bias (CB-PTS/D); and (4) confounding variables [48] (Appendix A). A score of either 0 or 1 was given to each criterion. Reason for limitations were defined for each criterion, if studies met one of the limitations they received a score of 0 for the criterion. The sum of given scores ranged from 0 to 4, with score 0–1 representing a low quality, score 2 a modest quality and score 3–4 a high quality [48]. 

The CASP checklist for qualitative studies is the most commonly used checklist for quality assessment in health-related research [50,51]. It comprises 10 items which were used to assess the overall quality of qualitative studies [52]. The results of the quality assessment were tabulated which is also seen in another review by Burke et al. (2022) [52] (Appendix A).

### 2.6. Data Extraction and Management

The following information was extracted from the included articles and reported using the Population, Exposure, Comparator and Outcomes (PECO) framework [53]: author, year of publication, country, population (i.e., number of participants, ethnicity, age and parity), exposure/comparator (mode of birth), outcomes (the measure used for assessing CB-PTS/D including the interval from birth to CB-PTS/D measurement (i.e., ≥6 months), and the main findings. The main findings report crude and adjusted associations expressed as odds ratio with 95% confidence intervals (OR (95% CI) calculated with univariable and multivariable logistic regression analyses. Additionally, we reported correlations (calculated with Spearman Rank- or Pearson correlation coefficient) and comparison of CB-PTS/D measurement scores (calculated with Student’s t-test or ANOVA).

## 3. Results

### 3.1. Search

The initial database searches identified 1319 articles of which 310 were identified through PubMed, 775 through Embase and 243 through CINAHL. Out of those articles, the total of 783 remained after removing the duplicates and articles written in languages other than English. After screening the titles and abstracts of the articles against our inclusion/exclusion criteria, a total of 136 articles were deemed eligible. After evaluating the full-texts of those articles, 28 articles were included in this integrative review. The detailed screening and selection process is outlined in a PRISMA flowchart (Figure 1).

### 3.2. Study Characteristics

A total of 28 studies were included in this review. Out of the included studies, 13 were longitudinal prospective cohort studies, 12 cross-sectional studies, one was a retrospective cohort study and two were qualitative studies. The included studies were published between 2004 and 2022, of which 15 studies were published after 2018 (Figure 2). They originated from 11 countries including the United Kingdom, Croatia, France, Netherlands, Turkey, Germany, Spain, United States of America, Australia, Sri Lanka, and New Zealand. 

The sample size ranged from six to 4509. In total, 26 studies reported the prevalence of CB-PTS/D, which ranged from 0.7% to 42%. 

Regarding the reported mode of birth in the included studies, four studies examined one mode of birth, i.e., vaginal birth [36,54,55] or emergency caesarean section (emergency CS) [56], whereas two studies examined two modes of birth, i.e., vaginal birth and caesarean section (CS) [22,57]. Six studies differentiated between vaginal birth, operative vaginal birth and CS [37,38,58,59,60,61], whereas three studies differentiated between vaginal birth, elective caesarean section (elective CS) and emergency CS [62,63,64]. Eleven studies distinguished between vaginal birth, operative vaginal birth, elective CS and emergency CS [18,65,66,67,68,69,70,71,72,73]. The two qualitative studies described women’s experiences after vaginal birth and CS [1,9]. 

In the selected studies, several questionnaires were used to screen for CB-PTS/D. Some were based on the DSM-IV criteria, including the Posttraumatic Diagnostic Scale (PDS) [9,22,38,57,59], PTSD Symptom Scale—self report (PSS-SR) [4,18,68], Traumatic Event Scale (TES) [36,60,61,66], Impact of Event Scale (IES) [36,58,67], Posttraumatic Stress Disorder Checklist (PCL) [65], Post-traumatic checklist scale (PCLS) [56], and Primary Care PTSD Screen (PC-PTSD-IV) [70]. The City Birth Trauma Scale (CityBits) [62,64,67,69,74], PTSD-short scale [54] and Perinatal PTSD Questionnaire (PPQ) [37,55,73,75] were based on the DSM-V criteria. 

Five studies administered the questionnaires during an interview with assistance from a healthcare professional [4,9,54,58,62], while 20 studies administered self-report questionnaires sent by postal mail, e-mail or online survey [18,22,36,37,38,55,56,58,59,60,61,64,65,66,67,68,69,70,73,75].

The data collection time ranged from 6 months to 5 years postpartum in the quantitative studies, while it ranged from 6 months until 18 years after childbirth in the qualitative studies. For the detailed information about the included studies see Table 1. In total twelve studies examined (crude or adjusted) associations between mode of birth and CB-PTS/D [22,36,37,38,58,59,61,66,70,72,73,75] (Figure 3).

### 3.3. Quality of the Studies

The majority of the included quantitative studies were of modest to high quality (Table 2). All qualitative studies met most of the criteria of high methodological quality, except for the criterion related to the description of the role of researcher [1,9]. Detailed quality assessment of the qualitative studies can be found in the Appendix A.

### 3.4. Findings Quantitative Studies

#### 3.4.1. Vaginal Birth

Compared with uncomplicated spontaneous vaginal births, vaginal births with intrapartum interventions and complications were associated with CB-PTS/D. One study found several intrapartum interventions such as administration of an enema, Kristeller maneuver, requirement to give birth in a supine position, artificial rupture of membranes without consent and repeated vaginal examination by different people during vaginal birth to be associated with CB-PTS/D [55]. However, no association was found if the labouring women were informed about those practices in advance and gave informed consent [55].

Moreover, the importance of the place of birth has been emphasized: compared with home births, women who gave birth in the hospital experienced CB-PTS/D more frequently [18,58]. One study focusing on vaginal birth, found that at 3 months post partum PTSD symptoms occurred particularly in women who reported lower physical labour comfort (individual satisfaction). However, lower physical labour comfort had no impact on PTSD symptoms at 6 months postpartum. [54]. One study reported that, although women with spontaneous vaginal birth versus those with other modes of birth, did not experience higher levels of PTSD symptoms, there were still individual women with spontaneous vaginal births who experienced severe PTSD symptoms 6 months postpartum [38].

#### 3.4.2. Operative Vaginal Birth

Eight studies indicated that operative vaginal birth was associated with CB-PTS/D, compared with spontaneous vaginal birth [36,37,38,58,66,70,72,75]. Four of those studies found an association between operative vaginal birth and CB-PTS/D in the univariate analyses, but this association was no longer significant after controlling for confounders such as social behavioral variables [36,66,70,75]. Nevertheless, in other studies the association between operative vaginal birth and CB-PTS/D remained significant even after controlling for confounders [37,38,58,72]. No association was found in one study [73]. One study reported a weak correlation between operative vaginal birth and CB-PTS/D in comparison with spontaneous vaginal birth [58]. Two studies reported statistically significantly higher symptom levels in women after operative vaginal birth compared to elective CS and vaginal birth [67,74] and three studies did not observe a significant effect on symptom scores [4,59,68].

#### 3.4.3. Caesarean Section (CS) (Not Differentiated into Elective and Emergency CS)

Three studies observed associations between CS (not differentiated into elective or emergency) and CB-PTS/D [37,38,59], as compared with vaginal birth Accompanying medical procedures for CS, e.g., general anesthesia, predicted specifically CB-PTS/D [61]. Two studies reported a very weak correlation between CS and CB-PTS/D symptom scores [38,57]. However, two studies did not observe an association [22] or correlation [57] between CS and CB-PTS/D. One study showed that women undergoing CS experienced less pain and had a less negative experience than women with vaginal birth; however, those women experienced more feelings of loneliness, losing control and hopelessness and feelings that correlated with CB-PTS/D at 1 year postpartum [60].

#### 3.4.4. Emergency Caesarean Section and Elective Caesarean Section

Eleven studies differentiated between emergency and elective CS in reporting their results. Three of them found an association between emergency CS and CB-PTS/D [70,72,75] compared with vaginal births. Two studies reported crude associations [70,75] whereas one study reported an adjusted association [72]. Seven studies reported higher CB-PTS/D symptom levels after emergency CS in comparison to other modes of birth [62,64,65,67,69,72,74], whereas two other studies did not find statistically significant elevated symptoms of PTSD in women after emergency CS compared to other modes of birth [58,68].

In two studies, emergency CS was not identified as a predictor of CB-PTS/D [66,73]. In addition, one study including only women with emergency CS, observed that women who experienced low satisfaction with emergency CS versus those experiencing high satisfaction with emergency CS were at higher risk of CB-PTS/D [56]. 

One study found that women with elective CS had higher levels of PTSD symptoms compared with other modes of birth [4]. Another study reported statistically significant higher symptom scores in women with elective CS compared to those with vaginal birth [69]. However, five studies found no association between elective CS and CB-PTS/D [66,70,72,73,75].

### 3.5. Findings Qualitative Studies

Two of the included studies were qualitative studies describing women’s experiences of CB-PTS/D. One study gathered information about CB-PTS/D by conducting a semi-structured interview with women (*n* = 6) [9]. The other study obtained data by sending out e-mails or postal mails, asking women (*n* = 38) to describe their experience of CB-PTS/D [1]. The qualitative studies suggest that CB-PTS/D can have severe and lasting effects on women’s lives such as struggling to survive each day while suffering from nightmares and flashbacks (i.e., at the moment of sexual penetration) which can lead to the avoidance of intimate relationships and suffering from anxiety, depression, anger and the feeling of isolation from other mothers [1,9]. 

## 4. Discussion

This is the first integrative review of the studies examining the impact of mode of birth (i.e., vaginal birth, operative vaginal birth, CS, elective CS and emergency CS) on CB-PTS/D beyond ≥6 months postpartum. The included studies were published from 2004 and 2022, the majority of them were published after 2018 which shows that this topic has attracted considerable clinical attention recently. Overall, the existing studies provide evidence that mode of birth represents a risk factor for CB-PTS/D in the long-term. More specifically, CB-PTS/D has been found to be associated with emergency CS and operative vaginal birth, but under certain circumstances also with elective CS and vaginal birth. Qualitative studies revealed that women were suffering from CB-PTS/D even 18 years after birth [9]. Several studies showed that the percentage of women suffering CB-PTS/D several months and years after childbirth (up to 5 years) remains high. The quantitative studies included in this review reported CB-PTS/D prevalence ranging from 0.7% to 42% in a time-frame of 6 months to 5 years postpartum. On a population level, CB-PTS/D prevalence of 2–9% has been reported in Western countries [4,5,6,7]. In comparison, postpartum depression, the most known psychiatric disorder in the postpartum period, has a prevalence of 17% worldwide [76].

The discrepancy between the CB-PTS/D prevalence found in the studies included in this systematic review and that reported in the literature [4,5,6,7] can be explained by the fact that one study included in this review reported a very high number of women with CB-PTS/D [54]. This study had however a relatively small sample size (*n* = 102) and used a recently developed measure to screen for PTSD that was not yet used in other studies [54]. Except for this study, the prevalence rate for CB-PTS/D reported in the studies included in this review is consistent with the range reported in the literature. It is not clear yet whether changes in the DSM criteria introduced in DSM-V result in changes in prevalence of CB-PTS/D [44]. Removal of the personal experience from the criterion A, which describes the traumatic event, might increase the number of births considered as traumatic [3,44], which could mean that more women might be diagnosed with CB-PTS/D.

Our integrative review contained a wide range of studies that varied in design, sample size and confounders controlled for. On comparison, it was found that cohort and longitudinal studies had more studies that examined associations between mode of birth and CB-PTS/D, than cross-sectional studies. Also, studies with lower sample size reported fewer associations than studies with a greater sample size. Additionally, some studies adjusted the associations between mode of birth and CB-PTS/D with mental health issues prior and during pregnancy (e.g., depression), the reported adjusted associations showed inconclusive results [22,58,61,70,73].

### 4.1. Mode of Birth

Compared with uncomplicated spontaneous vaginal births, vaginal births with intrapartum interventions and complications were more likely to be associated with CB-PTS/D [36,37,55,72]. As no association was found when women are informed and give consent before interventions [55], the lack of consent appears to be the precipitating factor and not the birth interventions as such [77].

Conditions under which operative vaginal birth is indicated are often extremely stressful due to the health condition of the mother or the fetus (such as maternal exhaustion or fetal distress) [78], but sometimes also due to lack of communication from healthcare professionals which may lead to a feeling of ‘non-care’, violation and powerlessness [4].

The studies included in this review and other published studies and systematic reviews that assessed short-term impact [15,79,80,81] show that operative vaginal birth is a risk factor for future CB-PTS/D. However, when operative vaginal birth was included in the models using multiple regression or hierarchical regression analyses along with other potential risk factors, it often did not remain a significant predictor of CB-PTS/D [36,66,70,75]. This could be explained by the lack of power due to a small study sample size in some studies [36], pre-existing conditions such as perinatal mental health issues [26], or social factors such as perceived safety during birth and partner support [66] which might be stronger risk factors for CB-PTS/D than mode of birth. Moreover, compassionate, and supportive care provided by the health care professionals may modify the impact of operative vaginal birth on CB-PTS/D [66].

Most of the studies indicated that, compared with vaginal birth, (elective and emergency) CS was associated with CB-PTS/D. Fetal distress and slow progress labour are the two main indications for emergency CS. While CS is a life-saving intervention the stress of the situation it occurs in can eventually increase the risk of CB-PTS/D [16,79,82]. Women who undergo elective CS may still develop PTS/D. Possible explanations for this may include fear of the procedure itself, a high-risk pregnancy, previous birth trauma or lack of support from the health care professionals during the procedure [83]. Also, some women tend to perceive CS as a personal failure [4,84,85]. In contrast, in countries with low medical resources and low CS rates, women who had to undergo medical interventions such as CS may consider themselves privileged to have been able to receive this (possibly) life-saving surgery [60]. In countries with high CS rates, CS is a more acceptable form of delivery [22,86] and this perception can minimize negative psychological consequences. This was potentially the case in the one study where operative vaginal birth and CS were not associated with CB-PTS/D [51]. Also, studies indicated that negative childbirth experiences of family members increase the risk of perceiving one’s own childbirth as traumatic [25,54]. It is important to acknowledge that it is not only support from, and optimal communication with health care professionals, but also the complex intersectional background of each woman that may play a role in the nature of her birth experience, i.e., in the extent to what she experiences operative birth as traumatic. Therefore, it is always important to consider the personal meaning of childbirth, family experiences, and cultural beliefs, in order to understand unique experience of childbirth and prevent trauma wherever possible.

Moreover, the subjective experience of childbirth which is influenced by social support from healthcare professionals plays an important role [21,87]. Personal perception of the extent to which the birthing mother is involved in the decision-making process (including respecting a birth plan) plays an important role in the quality of childbirth experience [37,75,88,89]. Lack of respect for the birth plan from health care professionals was associated with CB-PTS/D in two studies [37,55] as women’s wishes and expectations expressed in their birth plan were not in concordance with the experience they ultimately ended up having [37,55]. Obstetric violence such as verbal abuse from health providers during childbirth was reported to increase the risk of CB-PTS/D [55,68], as well as not being respected [55]. 

### 4.2. CB-PTS/D in the Long-Term

The majority of the cohort and longitudinal studies reported a slight decrease of CB-PTS/D over a period of time from baseline to 6 or 10 months [18,54,57,68,73]. These results were published between 2015 and 2022 suggesting that, given awareness regarding CB-PTS/D in both healthcare professionals and public in the last years, women were more likely to receive professional support and help. On the other hand, one longitudinal study and a meta-analysis reported, an increase in prevalence of CB-PTS/D over time from childbirth to 6 months, or from 6 months to 12 months postpartum [58,90]. It was explained that the increase was probably due to postnatal factors, such as sleep deprivation which might postpone CB-PTS/D symptoms onset and/or prevent remission [90]. One qualitative study reported that some women still suffered from CB-PTS/D 18 years postpartum [9]. They had flashbacks and avoided getting pregnant out of fear of reliving the traumatic birth [9]. An increase of the prevalence in CB-PTS/D diagnoses over time was observed in one prospective longitudinal cohort study [58]. This may be due to the development of more serious symptoms in women with partial CB-PTS/D (not full symptom spectrum measured by DSM-IV or V) or by additional traumatic events occurring after birth [83,91] which has also been observed in the studies assessing CB-PTS/D repeatedly up to 6 months [27,91]. However, it has been noted that if the symptoms last more than 6 months, spontaneous recovery is less likely, and the CB-PTS/D may require treatment [92]. One study suggested an association between chronic CB-PTS/D and poor social support after childbirth [26]. Another study stressed that the risk of developing chronic CB-PTS/D is linked with experiencing additional trauma since childbirth [22]. Importantly, when interpreting the course of CB-PTS/D in longitudinal studies, attrition rates should always be taken into consideration as the dropout of (vulnerable) participants can distort the results related to the longitudinal course and prevalence of CB-PTS/D. 

### 4.3. Strengths and Limitations

The key strengths of this integrative review include the focus on the long-term outcomes of mode of birth on CB-PTS/D. Careful identification of the studies that assessed CB-PTS/D using questionnaires or interviews based on DSM-IV and DSM-V criteria is also a strength. 

However, several limitations must be noted. First, methodological differences among the included studies, such as non-uniform categorizing of mode of birth (for instance: elective CS and emergency CS vs. CS not differentiated into subtypes), and different research aims and therefore varying emphasis on mode of birth in the individual studies made the comparison between studies challenging. Second, aside from selection bias and response bias in self-reporting questionnaires leading to over- or under-reporting of symptoms, assessing CB-PTS/D with different questionnaires in different studies resulted in varying prevalence rates of CB-PTS/D. Also, studies using clinical interviews versus questionnaires may report higher prevalence of CB-PTS/D [90]. Third, although all CB-PTS/D measures used in the included studies considered childbirth as the trigger for PTSD, some women already fulfilled the criteria for PTSD diagnosis during pregnancy. It is therefore unclear whether childbirth was indeed the trigger of their CB-PTS/D or whether it rather aggravated pre-existing or undiagnosed PTSD or new-onset of PTSD as a response to an event unrelated to childbirth which could coexist with a perceived traumatic childbirth [70].

### 4.4. Implications for Future Research and Recommendations

More longitudinal studies examining the impact of events occurring in pre-pregnancy, prenatal period on CB-PTS/D are needed. For example, invasive birth interventions such as vaginal examinations could increase the risk of developing CB-PTS/D in women with a history of sexual abuse [49]. According to the estimates by the WHO, 1 in 3 women globally are subjected to sexual violence [93], which means that many women are vulnerable to retraumatisation. Future longitudinal studies should be designed by incorporating the PECO framework: a population of pregnant women (without complications, women with obstetric risk factors or mental health disorders), exposure (birth interventions such as operative vaginal birth, caesarean section, induction or episiotomy), comparator (spontaneous vaginal birth), outcomes (measures of CB-PTS/D) assessed multiple times throughout a period of at least 1 year. Past traumatic life events (e.g., domestic violence), a history of mental health problems (e.g., anxiety or depression) and other risk factors for developing of aggravating CB-PTS/D should be controlled for along with protective factors promoting recovery from CB-PTS/D [22].

## 5. Conclusions

This integrative review summarizes the existing evidence about the long-term effects of mode of birth on CB-PTS/D. CB-PTS/D appears to be associated with emergency CS and operative vaginal birth. It is however of note that even vaginal birth may elicit severe CB-PTS/D. Therefore, screening for CB-PTS/D in the postnatal period could be of benefit although appropriate tools for and timing of such assessment is yet to be determined. Given its long-term adverse effects, CB-PTS/D should receive similar attention as postpartum depression among health care professionals and general population. Making new mothers familiar with CB-PTS/D might facilitate recognizing the symptoms and seeking professional support and help.

## Figures and Tables

**Figure 1 ijerph-19-08830-f001:**
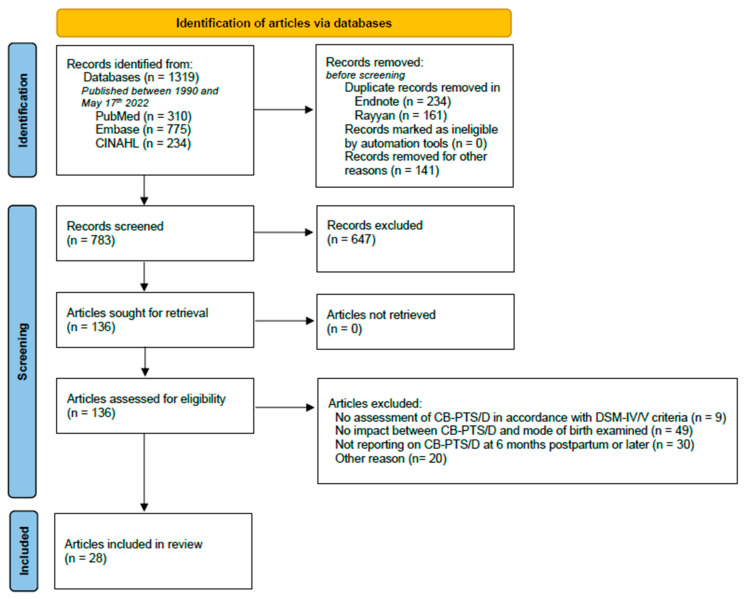
PRISMA flowchart illustrating the systematic database search and extraction of articles that met the inclusion criteria (MOB: mode of birth).

**Figure 2 ijerph-19-08830-f002:**
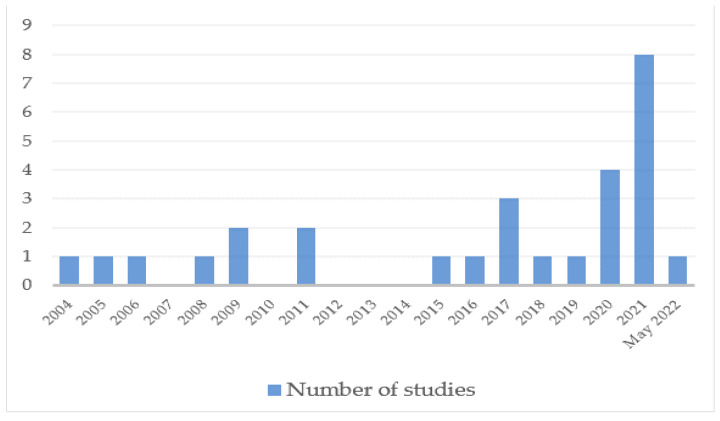
Number of studies per year.

**Figure 3 ijerph-19-08830-f003:**
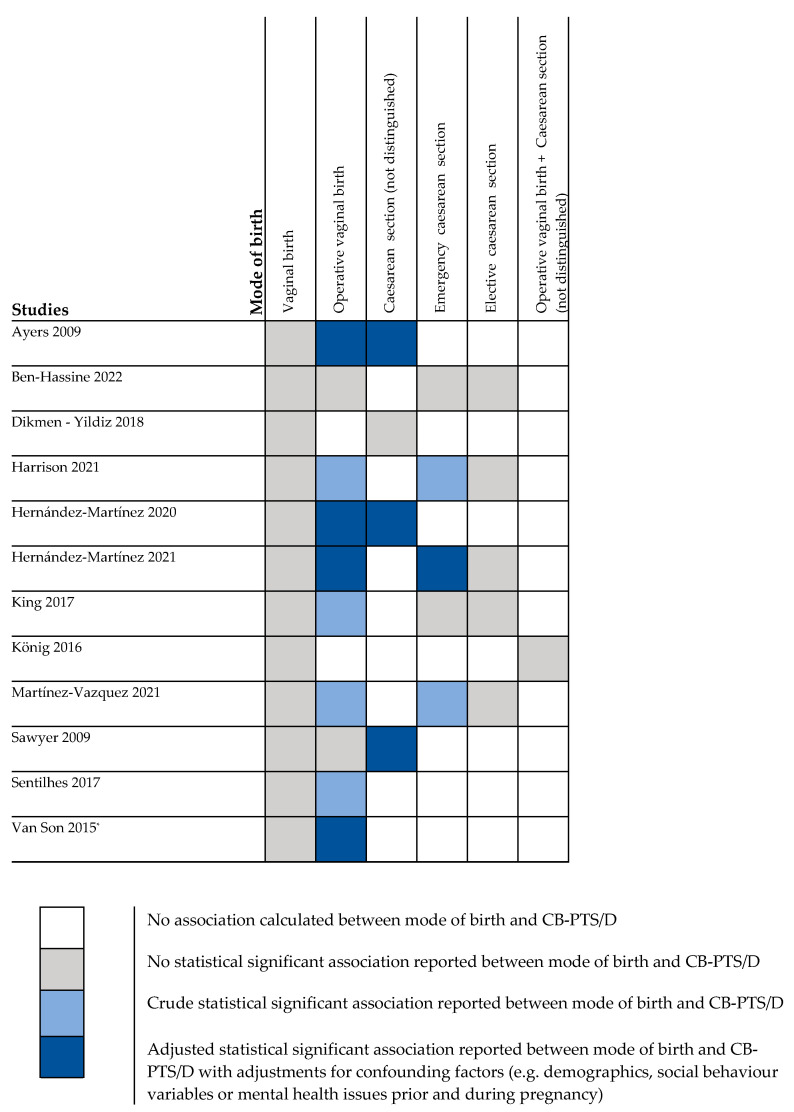
Associations between mode of birth and CB-PTS/D (≥6 months postpartum). * Significant association reported at 6 months postpartum not at 12 months postpartum.

**Table 1 ijerph-19-08830-t001:** Overview of included studies in the integrative review.

Authors, Year	Country	Study Design	Data Collection	Population Participants, n	Population Sample Characteristics			Exposure/ Comparison Mode of Birth ^a^, %	Outcomes Measure of CB-PTS/D ^b^	Postpartum Time of Data Collection ^c^	Main Findings including PTS/D Scores, Associations (Crude and Adjusted) and Correlations between Mode of Birth and PTS/D ^d^.
					Age in Years	Ethnicity, %	Parity, %				
Ayers et al., 2006 [9]	United Kingdom	Qualitative	Recruitment: via birth crisis network, response to a media article and word of mouth Time period: NR How: interview	6 women who experienced traumatic birth and reported CB-PTS/D in the first year after birth.	Range: 22–37 (age when given birth)	NR	Primiparous: 100	VB: 50 EmCS: 50	PDS DSM-IV Scores: 0–51 Cut off scores: 1–10 mild, 11–20 moderate, 21–35 moderate to severe, ≥36 severe	Between 7 m–18 y	2 women have CB-PTS/D, 3 women: residual symptoms, 1 woman recovered from CB-PTS/D
Ayers et al., 2009 [38]	United Kingdom	Cross-sectional (recruited from 4 cross-sectional internet studies, 3 longitudinal community studies)	Recruitment: 4 internet studies (relevant websites) and 3 community studies (antenatal clinics) Time period: NR How: Community: questionnaires per mail Internet: online questionnaire;	Total: 1297 Community 379 Internet 918	NR	White: Community: 80 Internet: 98	Community Primiparous: 57 Multiparous: 43 Internet Primiparous: 65 Multiparous: 35	Community VB: 61 OVB: 14 CS: 25 Internet VB: 51 OVB: 19 CS: 30	PDS Scores: 0–51 Cut off scores: 1–10 mild, 11–20 moderate, 21–35 moderate to severe, ≥36 severe PTSD cases: criteria A–F	Community Between 3 m–12 m Mean: 10.2 m (SD 4) Internet Mean: 12.8 m (SD 9)	Total prevalence CB-PTSD cases: Community: 2.5%, Internet: 21% Symptom clusters of PTSD: re-experiencing and avoidance, numbing and arousal Rho correlations between MOB and PTSD calculated with exploratory factor analysis Re-experiencing and avoidance symptoms (RAS) VB: 0.2 OVB: 0.4 CS: 0.5 Outliers with VB report high symptoms scores. Numbing and arousal symptoms (NAS) VB: 0.3 OVB: 0.4 CS: 0.6 Risk factors for CB-PTS/D PTSD symptoms: **OVB or CS: beta 0.28** (*p* ≤ 0.05). No OR/reported. PTSD cases: OVB or CS: not significant Associations between MOB and PTSD calculated with stepwise regression with backward method Confounders: PTSD symptoms: **age**, marital status, ethnic group, **history of sexual trauma** PTSD cases: **parity**, **numbing and arousal symptoms**, **re-experiencing and avoidance symptoms**, history of sexual trauma and mode of birth Predicting model: 60% of CB-PTSD cases were identified on the basis of parity (primiparous), MOB (OVB or CS), NAS and RAS, and interaction between sexual trauma and MOB.
Bayri Bingol et al., 2020 [62]	Turkey	Cross-sectional	Recruitment: At family health centres in Istanbul Time period: Aug–Oct 2018 How: Face to face interviews	481	Mean: 29 (SD 5)	NR	NR	VB: 62 ElCS: 30 EmCS: 8	CityBiTS DSM-V Scores: 0–60 Criteria A-H	6 m	Statistical analysis: student t-test, one way ANOVA, correlation, regression analyses Adjustment for confounders in regression analyses: confounders not reported Total prevalence CB-PTS/D: 8.5% CityBiTS scores mean (SD) **VB: 10.11 (9.62)** **ElCS: 12.32 (12.02)** **EmCS: 19.27 (14.48)** Statistical differences for different MOB were calculated with ANOVA (*p* ≤ 0.05)
Bayri Bingol et al., 2021 [64]	Turkey	Cross-sectional	Recruitment: At family health centres in Istanbul Time period: Aug–Oct 2018 How: self-report questionnaires	315	Mean: 29 (SD 5)	NR	NR	VB: 63 ElCS: 28 EmCS: 10	CityBiTS DSM-V Scores: 0–60 Criteria A-H	6 m	Total prevalence CB-PTS/D: 7.9% CityBiTs scores mean (SD) **VB: 9.69 (8.78)** **ElCS: 11.93 (11.77)** **EmCS: 17.93 (13.35)** Statistical differences for different MOB were calculated with ANOVA (*p* ≤ 0.05)
Beck et al., 2004 [1]	New Zealand United States Australia United Kingdom	Qualitative	Recruitment: via Internet Time period: 24 m How: description of experience of CB-PTSD detailed by e-mail or postal mail. Researcher asked further questions and for more examples	38 women with CB-PTSD diagnosed by a healthcare professional	Range: 25–44 Mean: 33	NR	Primiparous: 32 Multiparous: 68	VB: 55 CS: 45	Self-report CB-PTSD diagnosis	Between 6 w–14 y	42% women had subsequent pregnancies after traumatic birth (not all planned). Based on their traumatic birth experience, they were terrified of subsequent birth. 5% of them had a positive experience of their subsequent MOB, resulting in a recovery from CB-PTSD
Beck et al., 2011 [4]	USA	Longitudinal prospective cohort study	Recruitment: national survey Time period: T1: Jan–Feb 2006 T2: Jul–Aug 2006 How: online survey or telephone interview. Mothers were re-contacted and asked to complete a second questionnaire 6 m later	T1: 1573 Follow-up after 6 months: T2: 903	Range: 18–45	Caucasian: 63 Black: 12 Hispanic: 21 Asian: 4	T1 Primiparous: 33 Multiparous: 67 T2 Primiparous: 39 Multiparou: 61	T1 VB: 62 OVB: 6 CS: 32 (Primary CS: 16, Previous CS: 16) T2 VB: 62 OVB: 7 CS: 31 (Primary CS 17, Previous CS: 14)	PSS-SR DSM-IV Scores: 0–51 Cut off scores: ≥12 (some PTS symptoms) A-F Criteria for PTSD: based on DSM-IV	T1 Between 1 m–12 m Mean: 7.3 m (SD 3) T2 Between7 m–18 m	Total prevalence CB-PTS/D: 9% PSS-SR scores mean (SD) ElCS: 7.31 (9.72) Statistical differences between ElCS and not ElCS were assessed using T-test (*p* ≤ 0.05)
Ben-Hassine et al., 2022 [73]	France	Longitudinal prospective cohort study	Recruitment: hospital Time period: Mar 2018–May 2019 How: questionnaires by e-mail	T1: 256 women T2: 168 T3: 140	Mean: 32 (5)	NR	Primiparous: 49 Multiparous: 51	VB: 58 OVB: 18 ElCS: 13 EmCS: 11	PPQ DSM-V Cut off score: ≥6 Total score: 0–14	T1: after birth T2: 2 m T3: 6 m	Total prevalence CB-PTS/D: T2: 11,7%, T3: 10.5% Prediction models of CB-PTS/D (multiple hierarchical regression) No significant association found between MOB and CB-PTS/D. Confounders: Complications during pregnancy, depression during pregnancy, induced labour, use of epidural analgesia, pain during birth, complications during birth, **peritraumatic stress**, **emotions felt after birth**, interactions with health professionals
Deninotti et al., 2020 [56]	France	Cross-sectional	Recruitment: ads on several French social networking groups about CS on Facebook Time period: Mar–July 2016 How: online questionnaire	50 Women having CS under general anesthesia were excluded	Range: 18–35 Mean: 27 (SD 4)	NR	Primiparous: 76 Multiparous: 24	EmCS: 100	PCLS DSM-V cut-off score: ≥44	Between 1 m–24 m Mean: 10 m (SD 5)	Pearson correlation coefficient with CB-PTS/D Expressive suppression (inhibiting the expression of emotions) and low maternal satisfaction were very weakly correlated (R = 0.151) with CB-PTS/D. (*p* ≤ 0.05) Associations with CB-PTS/D Low maternal satisfaction: **B −0.38 SE 0.17 beta −0.32** (*p* ≤ 0.05) Emotion regulation strategies reappraisal/suppression: not significant Associations calculated with multiple regression analyses
Dikmen-Yildiz et al., 2017 [57]	Turkey	Longitudinal prospective cohort study	Recruitment: 3 state maternity hospitals Time period: May 2014–Jun 2015 How: T1: self-report questionnaire T2 + T3: telephone interview	T1: 950 T2: 858 T3: 829	Mean: 28 (SD 5)	NR	Primiparous: 40 Multiparous: 60	VB: 56 CS: 44	PDS DSM-IV Scores: 0–51 Cut off scores: 1–10 mild, 11–20 moderate, 21–35 moderate to severe, ≥36 severe	T1: 26–35 WOG T2: 4 w–6 w T3: 6 m	Total prevalence CB-PTS/D: at T2: 11.9%, T3: 9.2% (5.8% fulfilled all CB-PTS/D criteria already in pregnancy) Spearman Rank Correlation between MOB and PDS scores **T2: −0.08** (*p* ≤ 0.05) T3: not significant
Dikmen-Yildiz et al., 2018 [22]	Turkey	Longitudinal prospective cohort study	Recruitment: 3 state maternity hospitals Time period: May 2014–Jun 2015 How: T1: self-report questionnaires T2 and T3: telephone interview	T1: 895 T2: 287 T3: 279 T2 + T3 Only women with traumatic birth experience according to DSM-IV	No CB-PTS/D: Mean: 27 (SD 5) CB-PTS/D: Mean: 27 (SD 5)	NR	No CB-PTS/D: Primiparous: 59 Multiparous: 41 CB-PTS/D: Primiparous: 52 Multiparous: 48	Resilient: VB: 75 CS: 25 Recovered: VB: 59 CS: 41 Chronic: VB: 68 CS: 32 Delayed: VB: 23 CS: 77	PDS DSM-IV Resilient: (No CB-PTSD) Recovered: (CB-PTS/D at T2 but none at T3) Delayed: (none at T2 but T3) Chronic: (T2 + T3) Non-resilient: recovered+delayed+chronic Scores: 0–51 Cut off scores: 1–10 mild, 11–20 moderate, 21–35 moderate to severe, ≥36 severe	T1: 26–35 WOG T2: 4 w–6 w T3: 6 m	Total prevalence CB-PTS/D: Resilient: 61.9%, Recovered: 18.5%, Delayed: 5.8%, Chronic: 13.7% Proportion CS (%) Resilient 25%; Non-resilient 43% With Chi square statistical differences calculated between resilient and non-resilient women with CS (*p* < 0.01) Predictor of PDS scores: CS Not significant predictor; tested with bivariate logistic regression Predictor: CS (after exclusion of affective symptoms as covariates) (at T2): Recovered vs resilient, chronic vs resilient Not significant predictor Delayed vs resilient **b 1.73 SE 0.83 OR: 5.65 (95% CI 1.11–28.69)** Association calculated between delayed group with CS and PSD scores with multivariate logistic regression. (*p* ≤ 0.05) Resilient and non-resilient women with CS as a predictor of CB-PTS/D Not significant predictor calculated with multivariate logistic regression Delayed-CB-PTS/D: All women having preterm birth in the delayed group gave birth by CS which may contribute to the association between CS and delayed-CB-PTS/D. Confounders: Intra-partum complication, gestational age (preterm), postpartum complication, infant complication, traumatic event after birth, psychological help received, **satisfaction with health professionals**, affective symptoms in pregnancy (depression, anxiety), **affective symptoms at 4–6 weeks postpartum**, fear of birth symptoms at 4–6 weeks, social support at 4–6 weeks postpartum
Dobson et al., 2022 [74]	Australia	Cross- sectional	Recruitment: social media Time period: Mar–Apr 2020 How: NR	195	Range: 20–42 Mean: 32	Caucasian: 88 Asian: 2 Indian or Sri Lankan 0.5 Other: 7	Primiparous: 67 Multiparous: 33	VB: 47 OVB: 16 ElCS: 14 EmCS: 23	CityBiTS DSM-V Scores: 0–60 Criteria A–H	Between 1 m–12 m	Total prevalence CB-PTS/D: 8.2% CityBiTs scores mean (SD) Birth-related symptoms(distressing memories and avoiding thinking about the birth) VB: 3.88 (5.98) **OVB: 5.61 (7.38)** ElCS: 1.63 (4.14) **EmCS: 7.76 (7.59)** Statistical difference for OVB and EMCS in comparison to VB and ElCS calculated with ANOVA (*p* < 0.001) General symptoms (assessing negative cognitions/mood and hyperarousal) VB: 7.98 (8.18) OVB: 9.48 (8.30) ElCS: 6.41 (5.96) EmCS: 9.11 (7.25) No statistical difference for different MOB calculated with ANOVA. Total scale VB: 11.86 (12.07) OVB: 15.1 (14.57) ElCS: 8.04 (8.62) EmCS: 16.87 (13.35) Statistical difference for different MOB calculated with ANOVA (*p* = 0.007)
Gankanda et al., 2021 [68]	Sri Lanka	Longitudinal prospective cohort study	Recruitment: 4 field clinics in Horana MOH area Time period: NR How: interviewer administering checklist and self-administered questionnaires	T1: 225 T2: 214 T3: 211	Range: 15–42 Mean: 28	NR	Primiparous: 42 Multiparous:58	VB: 57 OVB: 2 ElCS: 36 EmCS: 4	PSS-SR DSM-IV Scores: 0–51 cut-off score: ≥13	T1: 1 m T2: 2 m T3: 6 m	Total prevalence CB-PTS/D (PSS-SR 13): 3.6% Incidence: T1: 2.7%, T2: 0.9%, T3: 0,5% Proportion CB-PTS/D: VB: 2.3% OVB: 0% ElCS: 4.9% EmCS: 10% No statistical difference between different MOB shown with Fisher Exact tests (*p* ≤ 0.05)
Haagen et al., 2015 [18]	Netherlands	Longitudinal prospective cohort study	Recruitment: midwife practices Time period: Sep 2001–Apr 2004 How: T1: interview with midwife and self-reported questionnaire T2: self-report questionnaires collected by midwife T3 + T4: via mail.	T1-T3: 348 T4: 284	Mean: 31	Caucasian: 100	Primiparous: 42 Multiparous: 58	VB: 84 (Home: 48 Hospital: 24) OVB: 8 ElCS: 3 EmCS: 5	PSS-SR DSM-IV Scores: 0–51 cut-off score: ≥13 PSS-SR symptom criteria: score above 18 and 1 symptom of criterion B, 3 symptoms of criterion C, 2 symptoms of criterion D PTSD diagnosis: fulfilment of DSM-IV A-E criteria	T1: 18 WOG T2: 1st wk T3: 3 m T4: 10 m	Total prevalence: PSS-SR symptom criteria + A criterion T3: 1.7% T4: 0.70% Total prevalence: CB-PTSD diagnosis T3: 0.57% T4: 0.35% Postpartum model predicting CB-PTS/D severity at T4: MOB (ordinal variable in which each MOB becomes more invasive) predicted (β = 0.15, *p* ≤ 0.05) negative emotional responses, which had an indirect effect (β = 0.14, SE = 0.058, *p* ≤ 0.05) on CB-PTS/D severity via somatoform dissociation. This model accounted for 24% of CB-PTS/D symptom variability.
Harrison et al., 2021 [70]	England	Cross-sectional	Recruitment: selected randomly by Office for National Statistics using birth registration records Time period: Oct 2017–two-week interval How: on paper, online or by telephone with an interpreter if required	4509	Range: 29–36 Mean: 32	White-British: 76 Other: 24	No CB-PTS/D: Primiparous: NR Multiparous:NR With CB-PTS/D: Primiparous: 52% Multiparous: 48%	VB: NR OVB: NR ElCS: NR EmCS: NR	PC-PTSD-IV Scores: 0–4 cut-off score: ≥3	6 m	Total prevalence: CB-PTS/D: 2.5%, the symptom reported most frequently by the women with CB-PTS/D was re-experiencing; hyperarousal was reported least frequently. Prevalence (n), (%) VB: 2344 1.8% OVB: 602 4.2% ElCS: 586 3.1% EmCS: 617 4.8% Crude OR 95% CI calculated with univariate logistic regression: VB: (ref) OVB: **2.44 (95% CI 1.25–4.76)** ElCS: 1.75 (95% CI 0.93–3.29) EmCS: **2.81 (95% CI 1.55–5.09)** Adjusted OR (95% CI) calculated with multivariate logistic regression: None of the MOB was associated with CB-PTS/D Confounders: **higher level of deprivation**, multiple birth, **not having a healthcare professional to talk to about sensitive** **issues during pregnancy**, experiencing childbirth worse than expected, **the baby admitted to the neonatal intensive care unit**, **living without a partner**, a neutral or mixed reaction to pregnancy, **anxiety during pregnancy**, depression during pregnancy, having a pregnancy affected by long-term health problems, **pregnancy-specific health problems**, **lower satisfaction with birth**
Hernández-Martínez et al., 2020 [37]	Spain	Cross-sectional	Recruitment: via different women and midwives associations Time period: NR How: Online questionnaire	1531 women at least 1 yr postpartum	<35 y: 42% ≥35 y: 58%	NR	Primiparous: 53 Multiparous: 47	VB: 57 OVB: 18 CS: 25	PPQ DSM-V Scores: 0–56 cut-off score: ≥19	Between 1 yr–5 yr	Total prevalence CB-PTS/D Total: 7.2%, 1–3 y: 8.1%, 4–5 y: 5.9% Adjusted OR (95%CI) calculated with multivariate logistic regression: VB (ref) OVB: **3.32 (95% CI 1.73−3.39)** CS: **4.80 (95% CI 2.51–9.15)** Confounders: Mother’s age, **parity**, **birth plan respected**, **use of epidural/rachianaesthesia**, **fundal pressure**, **mode of birth**, **third/fourth degree perineal tears**, **skin to skin**, **postpartum time**
Hernández-Martínez et al., 2021 [72]	Spain	Observational retrospective cohort study	Recruitment: NR Time period: 2018–2019 How: medical records	Derivation cohort (DC): 1752 Validation cohort (VC): 875	DC: ≤35: 43% >35: 57% VC: ≤35: 43% >35: 57%	DC: Spanish: 96 Other: 4.1 VC: Spanish: 97 Other: 3	DC: Primiparous: 69 Multiparous: 31 VC: Primiparous: 66 Multiparous: 34	DC: VB: 57 OVB: 19 ElCS: 7 EmCS: 17 VC: VB: 61 OVB: 17 ElCS: 7 EmCS: 15	PPQ DSM-V Scores: 0–56 cut-off score: ≥19	Between 1 m–18 m Mean: 15.7 m (SD 1.77)	Total prevalence CB-PTS/D: DC: 14.2%, VC: 10.9% Proportion ≥ 19 points VB: 84 (8.4%) OVB: 52 (16.0%) ElCS: 20 (15.5%) EmCS: 92 (30.5%) Statistical differences for different MOB calculated with T-test (*p* ≤ 0.05) Prediction models of CB-PTS/D Model A: clinical criteria VB: (ref) OVB: **OR 1.62 (95% CI 1.10–2.41)** ElCS: not significant EmCS: **OR 3.07 (95% CI 1.96–49.0)** Model B: clinical criteria + maternal perceptions of partner support and treatment received by healthcare professionals VB: (ref) OVB: not significant ElCS: not significant **EmCS: OR 2.29 (95% CI 1.56–3.35)** Predictive models for CB-PTS/D were created using multivariate binary logistic regression. Confounders: Model A: **initiate skin-to-skin contact**, **admission of the newborn to care unit**, perineal tear type 1–2, **perineal tear type 3–4**, **infant feeding on discharge (mixed, artificial)**, **hospital readmission** Model B: **admission of newborn to care unit**, **infant feeding on discharge** (mixed, **artificial**), **hospital readmission**, **partner’s perception of support**, **perception of respect by professionals**
King et al., 2017 [66]	United Kingdom	Cross-sectional	Recruitment: through online and paper sources such as advertisements Time period: Dec 2013–May 2014 How: Questionnaires hosted on a survey website	157	Range: 18–44 Mean: 30	White: 94 Black African: 1 Other: 5	NR	VB: 67.5 OVB: 18.5 ElCS: 4 EmCS: 10	TES DSM-IV	Between 1 m–12 m Mean: 6.5 m	Total prevalence CB-PTS/D: 5.7% Predictors of CB-PTS/D **OVB: beta 0.16 SE 1.49 (*p* ≤ 0.05)** ElCS: Not significant EmCS: Not significant Predictors were calculated with simultaneous multiple regression analysis. Confounders: **Perceived safety**, **positive social interaction**, single, **negative cognitions of self**, **rumination**, **numbing**, **deficits in intentional recall**, **negative appraisals of memory deficits**
König et al., 2016 [61]	Germany	Longitudinal prospective cohort study	Recruitment: Maternity wards in five hospitals Time period: May 2013–April 2014, 4–6 w per hospital How: T1: questionnaire in hospital T2 + T3: questionnaires by mail	T1: 353 T2: 263 T3: 227	Mean: 33 (SD 5)	NR	Primiparous: 41 Multiparous: 59	VB: 22 CS: 33 OVB: 45 Episiotomy unknown Medical interventions (CS, OVB or episiotomy): 78	TES DSM-IV PTSD diagnosis: all criteria of DSM-IV A-F	T1: shortly after birth T2: 6 w T3: 1 yr	Correlation and prediction models of TES at T3 Correlation and predictors were calculated with Pearson Correlation coefficient and multivariate binary logistic regression. Number of medical interventions (OVB, CS) r: 0.19 *p* ≤ 0.05 (very weak correlation) Model 1: Not significant Model 2: Not significant General anaesthesia r: 0.27 *p* = 0.0001 (weak correlation) **Model 1: B: 3.55, SE B: 1.70, beta: 0.14 (*p* ≤ 0.05)** (authors indicate that general anaesthesia acts as a dummy variable for more difficult CS.) Confounders: Model 1: foreign language spoken, **antidepressant in last 10 years**, **episiotomy**, number of medical interventions, **general anaesthesia**, **wijma delivery experience questionnaire** Model 2: confounders of Model 1 + **TES at T2**, Edinburgh postnatal depression scale at T2, general health questionnaire at T2, satisfaction with physical state at T2, **pain at T2**
König et al., 2019 [60]	Germany	Longitudinal prospective cohort study	Recruitment: Maternity wards in five hospitals Time period: May 2013–April 2014, 4–6 w per hospital How: T1: questionnaire in hospital T2 + T3: questionnaires by mail	T1: 353 T2: 263 T3: 227	Mean: 33 (SD 5)	NR	Primiparous: 41 Multiparous: 59	VB: 22 CS: 33 OVB: 45 Medical intervention (CS, OVB or episiotomy): 78	TES DSM-IV PTSD diagnosis: all criteria of DSM-IV A-F W-DEQ part B (subjective experience of childbirth)	T1: shortly after birth T2: 6 w T3: 1 yr	TES: after T2 + T3: factors ‘lack of self-efficacy, fear, and negative experience’ were most important and had the strongest correlations with CB-PTS/D W-DEQ: CS scored higher on ‘loneliness and fear’ than VB. CS scored lower on ‘negative experience’ than VB and OVB probably due to less ‘pain’. In total: VB scored lower than CS in general, while women with OVB did not differ significantly from the women with other MOB
Leeds et al., 2008 [65]	United Kingdom	Cross-sectional	Recruitment: alternate randomization of 479 women who gave birth at district general hospital Time period: Oct 2003–Mar 2004 How: questionnaires by mail	102	Non symptomatic (NS) Mean: 30 Partially symptomatic (PS) Mean: 31 Fully symptomatic (FS) Mean: 26	NR	NS: Primiparous: 40 Multiparous: 60 PS en FS: Primiparous: 50 Multiparous: 50	NS VB: 60 OVB: 9 ElCS: 21 EmCS: 10 PS VB: 45 OVB: 20 ElCS: 5 EmCS: 30 FS VB: 50 EmCS: 50	PPQ DSM-V Scores: 0–56 cut-off score ≥ 19 PS significant in one area (criterion B, C or D) FS significant in criterion B, C and D	Between 6 m–12 m Mean: 9.5 m	Total prevalence CB-PTS/D: FS: 3.9% PS: 19.6% Proportion CB-PTS/D (%) FS VB: 50% OVB: 0% ElCS:0% EmCS: 50% PS VB: 45% OV: 20% ELCS: 5% EmCS: 30% NS VB: 60.2% OVB: 8.9% ElCS: 20.5% EmCS: 10.2% 33,3% of PS and FS delivered by EmCS in comparison to 10% of the NS No statistical test conducted.
Martínez-Vazquez et al., 2021, 10 [75]	Spain	Cross-sectional	Recruitment: public or private hospitals or at home Time period: 2019 How: through online questionnaire	1301 Women with previous psychiatric history and history of PTSD excluded.	Mean: 36 (SD 4)	NR	Primiparous: 71 Multiparous 29	VB: 57 OVB: 17 ElCS: 8 EmCS: 18	PPQ DSM-V Scores: 0–56 cut-off score: ≥19	Between 12 m–38 m	Total prevalence CB-PTS/D: 13.1% Association between MOB and PPQ scores VB 1 (ref) **OVB: OR 2.20 (95% CI 1.42–3.39)** ElCS: Not significant **EmCS: OR 3.57 (95% CI 2.41–5.28)** Statistical significant association calculated with bivariate analysis (crude OR (95%CI) (*p* ≤ 0.05)
Martínez-Vazquez et al., 2021, 04 [55]	Spain	Cross-sectional	Recruitment: via midwives Time period: Sep–Dec 2019 How: online questionnaire	839	Mean: 36	NR	Primiparous: 51 Multiparous: 49	VB: 100	PPQ DSM-V Scores: 0–56 cut-off score: ≥19	Between 1 m–12 m Mean: 7.17 m	Total prevalence CB-PTS/D: 8.1% administration of an enema **aOR 7.01 (95% CI 2.14–23.01)** being required to stay lying down throughout the labor and birth **aOR 5.75 (95% CI 3.25–10.19)** artificial amniorrhexis without consent **aOR: 2.28 (95% CI: 1.31–3.97)** administration of synthetic oxytocin without consent **aOR 2.18 (95% CI 1.26–3.77)** fundal pressure during pushing **aOR 3.14 (95% CI 1.72–5.73)** repeated vaginal examinations performed by different people **aOR 4.84 (95% CI 2.77–8.47)** manual removal of the placenta without anesthesia **aOR 3.45 (95% CI 1.81–6.58)**.
Nakic Radoš et al., 2020 [67]	Croatia	Cross-sectional	Recruitment: online: Facebook group postings, shared via personal contacts. Time period: Nov 2018–Dec 2018 How: online questionnaires	603	Range: 20–47 Mean: 31 (SD 5)	NR	Primiparous: 61 Multiparous: 39	VB: 75OVB: 2ElCS: 8 EmCS: 15	City BiTS DSM-V Scores: 0–60 IES-R DSM-IV Scores: 0–88 Diagnosis: criteria A-H met	Between 1 m–12 m Mean: 6.1 m (SD 3)	Total prevalence Criterion A: 31.18% Total prevalence CB-PTSD: 11.77% 78.3% reported onset of CB-PTS/D within the first 6 months postpartum 47.7% reported having symptoms for more than 3 months CityBiTs scores mean (SD) Birth-related symptoms (distressing memories and avoiding thinking about the birth) VB: 4.85 (6.71) **OVB: 11.33 (9.18)** ElCS: 3.45 (5.66) **EmCS: 8.65 (8.14)** Statistical differences calculated for different MOB with ANOVA (*p* ≤ 0.05) General symptoms (assessing negative cognitions/mood and hyperarousal) VB: 8.80 (8.16) OVB: 12.42 (8.37) ElCS: 10.11 (8.94) EmCS: 10.36 (8.11) No statistical differences were found for different MOB with ANOVA (*p* = 0.15) Total scale VB: 13.65 (12.51) **OVB: 23.75 (14.78)** ElCS: 13.55 (12.96) **EmCS: 19.01 (14.16)** Statistical differences calculated for different MOB with ANOVA (*p* ≤ 0.05)
Sawyer et al., 2009 [59]	United Kingdom	Cross-sectional	Recruitment: via the internet on different websites. Time period: NR How: online questionnaire	219	Range: 18 -42 Mean: 28	White: 97Afro Caribbean: 0.5 Indian/Pakistani: 1 Other: 1.5	Primiparous:65 Multiparous: 35	VB: 63 OVB: 11 CS: 26	PDS DSM-IV Scores: 0–51 Diagnosis: criteria A-F met Symptom severity score: Cut offs: 1–10 mild, 11–20 moderate, 21–35 moderate to severe, ≥36 severe	Between 1 m–36 m Mean: 11 m (SD 7)	A difference in CB-PTS/D across MOB was shown With ANOVA statistical differences calculated for different MOB (*p* ≤ 0.05) CB-PTS/D was significantly higher if women had a CS compared to VB or OVB; shown by post-hoc comparison using Games-Howell (*p* ≤ 0.05) Predictors of CB-PTS/D: MOB Step 1: **B: 0.11 SEB: 0.04, beta: 0.19** Step 2: Not significant Hierarchical multiple regression analysis showed that model 1 accounted for 46.4% of the variance in CB-PTS/D scores (*p* ≤ 0.05) Confounders: Pain, approach, **avoidant**, **external control**, **internal control**, support
Sentilhes et al., 2017 [36]	France	Longitudinal prospective cohort study	Recruitment: 5 hospitals Time period: Jan 2010–Jan 2011 How: postal questionnaire	549 women who were previously enrolled in RCT (TRACOR trial) at ≥35 WOG	Age groups: <25: 11% 25–34: 73% >35: 16%	NR	Primiparous: 46 Multiparous: 54	VB: 88 OVB: 12	TES DSM-IV PTSD diagnosis: all criteria of DSM-IV A-F IES Cut-off score ≥26: very serious PTS/D ≥19: clinically significant	T1: 2 d T2: 12 m	Total prevalence CB-PTSD diagnosis (TES): 4.2% IES score: >20: 8.6, >26: 40, >31: 2.5 Univariate analysis of factors associated with CB-PTSD diagnosis at T2 TES: OVB: **OR 2.4 (95%CI 1.0–6.0)** IES: OVB: **OR 3.8 (95%CI 1.5–9.7)** Multivariate analysis of factors associated with CB-PTSD diagnosis at T2 TES/IES: OVB: Not significant Confounders: TES: **previous abortion**, **previous postpartum hemorrhage**, hospitalization during pregnancy, instrumental delivery, episiotomy IES score > 26 **Previous abortion**, labor > 6 h
Türkmen et al., 2020 [54]	Turkey	Longitudinal prospective cohort study	Recruitment: hospital in the delivery room Time period: Jun 2019–Feb 2020 How: self-report questionnaires in delivery room, then face-to-face interviews	102 pregnant women who planned a vaginal delivery	Mean: 26	NR	Primiparous: 60 Multiparous: 40	VB: 100	PTSD- Short scale DSM-V Scores: 9–45, Cut-off score: ≥24	T1: 0 d T2: 4 w T3: 3 m T4: 6 m	Total prevalence CB-PTS/D: T3: 52.9%, T4: 42.2% Lower physical labour comfort at 3 m was associated with CB-PTS/D, but was not associated at 6 m with CB-PTS/D. CB-PTS/D was significantly related to subjective recall of labor experience. As traumatic childbirth experience increases, CB-PTS/D increases.
Van Son et al., 2005 [58]	Netherlands	Longitudinal prospective cohort study	Recruitment: At midwife or obstetrician appointment Time period: NR How: interview	T1: 248 T2-T4: NR	Range: 19–43 Mean: 31 (SD 4)	NR	Primiparous: 43 Multiparous: 57	VB home: 26 VB hospital: 35 CS: 11 OVB: 9	IES Scores 8–25: warrants serious clinical attention ≥26: very serious symptoms	T1: 34 WOG T2: 3 m T3: 6 m T4: 12 m	Total prevalence CB-PTS/D: IES ≥ 26 score: T2: 8.1%, T3: 3%, T4: 5%. Total prevalence CB-PTS/D: IES (8–25): T2: 38%, T3: 42%, T4: 48% IES scores mean (SD) at T2, T3, T4 VB at home: 6.9 (7.9), 6.5 (6.4), 7.6 (7.3) VB in hospital: 10.4 (11.0), 7.5 (8.2), 9.4 (9.5) CS: 10.0 (9.3), 9.8 (9.8), 10.0 (9.6) OVB: 14.8 (10.0), 10.0 (8.9), 13.7 (8.0) Linear trend in IES scores along the severity of MOB, the mean scores did not differ statistically significantly. Severity of MOB and IES scores showed a Pearson correlation coefficient of 0.35. Regression analyses regarding percentage on IES in relation to MOB showed only a difference between VB at home and OVB (forceps) (*p* ≤ 0.05). CB-PTS/D at 6 months was not affected by perinatal dissociation (*p* > 0.05), but by earlier PTS, mode of delivery, and depression during life and depressive symptoms at 6 months postpartum (all *p* < 0.05) At 12 months CB PTS/D was affected by perinatal dissociation (*p* < 0.05) and indirectly by type of delivery, pain, and social support/information during delivery (all *p* < 0.05).
Weigl et al., 2021 [69]	Germany	Cross-sectional	Recruitment:community sample online (social media, forums) Time period: Feb–Apr 2020 How: online questionnaire	1072	Range: 18–44 Mean: 31	NR	Primiparous: 60 Multiparous: 40	VB: 69 OVB: 8 ElCS: 7 EmCS: 16	City BiTS DSM-V Scores: 0–60 Diagnosis: Criteria A–H IES-R + PCL-5 as validation	Mean: 6 m (SD 3.3)	Total prevalence CB-PTS/D: 2.6% CityBiTs scores mean (SD) Birth-related symptoms (distressing memories and avoiding thinking about the birth) **VB: 2.73 (4.82)** **OVB: 6.51 (7.39)** **ElCS: 5.61 (6.57)** **EmCS: 7.94 (7.53)** Statistically significant differences between VB and all MOB and difference between ElCS and EmCS calculated with ANOVA (*p* ≤ 0.05) General symptoms (assessing negative cognitions/mood and hyperarousal) VB: 5.61 (6.27) OVB: 5.74 (6.03) ElCS: 7.24 (6.92) **EmCS: 7.89 (7.15)** Statistically significant difference between VB and EmCS calculated with ANOVA (*p* ≤ 0.05) Total scale **VB: 8.35 (9.28)** **OVB: 12.26 (12.91)** **ElCS: 12.85 (12.11)** **EmCS: 15.83 (13.15)** Statistically significant differences between VB and all MOB calculated with ANOVA (*p* ≤ 0.05)

NR: not reported. ^a,d^ VB: vaginal birth, OVB: operative vaginal birth, CS: caesarean section, EmCS: Emergency caesarean section, ElCS: Elective caesarean section, MOB: mode of birth. ^b^ PDS: posttraumatic diagnostic scale, CityBits: city birth trauma scale, PSS-SR: PTSD symptom scale—self report, TES: Traumatic Event Scale, IES: Impact of Event Scale, PCL: Posttraumatic Stress Disorder Checklist, PCLS: Post-traumatic checklist scale, PC-PTSD-IV: Primary Care PTSD Screen, CityBits: The City Birth Trauma Scale, PTSD-short scale, PPQ: Perinatal PTSD Questionnaire, W-DEQ part B: wijma delivery expectancy/experience questionnaire. ^c^ WOG: week of gestation. ^d^ Bold text indicates statistical significance.

**Table 2 ijerph-19-08830-t002:** Quality assessment of the included studies using the Critical Appraisal Skills Programme (CASP) methodology.

Studies	Selection Bias Criterion 1	Measurement Bias		Confounding Criterion 4	Total (0–4)
		Exposure Criterion 2	Outcome Criterion 3		
*Quantitative studies*					
Ayers 2009 [38]	1	0	1	1	3
Bayri Bingol 2020 [62]	1	0	1	1	3
Bayri Bingol 2021 [64]	1	0	1	1	3
Beck 2011 [4]	1	0	1	0 b	2
Ben-Hassine 2022 [73]	0 c	1	1	0 a	2
Deninotti 2020 [56]	0 a	1	1	0 a,b	2
Dikmen-Yildiz 2017 [57]	1	0	1	1	3
Dikmen-Yildiz 2018 [22]	1	0	1	1	3
Dobson 2022 [74]	0	1	1	0 a	2
Gankanda 2021 [68]	1	1	1	0 a	3
Haagen 2015 [18]	1	1	1	0 a	3
Harrison 2021 [70]	0 a,c	1	1	0 a	2
Hernández-Martínez 2020 [37]	0	0	1	0 b	1
Hernández-Martínez 2021 [72]	1	1	1	0 b	3
King 2017 [66]	0 a	1	1	1	3
König 2016 [61]	1	0	0	0 a,b	1
König 2019 [60]	1	0	0	0 a,b	1
Leeds 2008 [65]	0	1	1	1	3
Martínez-Vasquez 2021, 04 [55]	1	1	1	0 a,b	3
Martínez-Vazquez 2021, 10 [75]	1	1	1	0 b	3
Nakic Radoš 2020 [67]	0 a	1	1	0 a	2
Sawyer 2007 [59]	0 a	0	1	1	2
Sentilhes 2017 [36]	0	1	1	0 a,b	2
Türkmen 2020 [54]	0 b	0	1	0 a,b	1
Van Son 2005 [58]	1	0	1	0 a	2
Weigl 2021 [69]	0 a	1	1	0 b	2
*Qualitative studies*					
Ayers 2006 [9]	Study meets most criteria of high methodological quality according to CASP, except for a limited description of recruitment and role of researcher.				
Beck 2004 [1]	Study meets most criteria of high methodological quality according to CASP, except for lacking a description of the role of researcher.				

Reason for limitations: Criterion 1: a: Recruitment population does not reflect true population, b: No description of recruitment, c: Low response rate Criterion 2: No distinction between emergency and elective CS or between (operative) vaginal birth Criterion 3: No validated questionnaire Criterion 4: Confounders: (a) No previous trauma (e.g., sexual abuse) or (b) mental health (e.g., depression).

## Data Availability

Not applicable.

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
