# Peer review of "The Impact of Mode of Birth on Childbirth-Related Post Traumatic Stress Symptoms beyond 6 Months Postpartum: An Integrative Review"

_ijerph, 2022, doi:10.3390/ijerph19148830_

Round 1

Reviewer 1 Report

Lines 18 and 19: Knowledge on the risk factors .... is researched thoroughly. Can the authors clarify this statement ie do you mean this study is researching the subject thoroughly, or are you stating the literature has identified the risk factors for CB-PTSD and that the risk factors identified relate to the likelihood of developing symptoms only in the short-term? 

I note the criteria for the diagnosis of PTSD have changed with the introduction of the DSM-V. Could the authors comment as to whether this affects the prevalence of CB-PTSD?

The authors indicate that PTSD symptoms or the diagnosis vary between 0.7% and 42% in the studies included in the analysis and contrast this with the results of western prevalence studies that suggest 2-9%. Is there a reason why the marked discrepancy between these figures?

The authors have not commented on issues related to the use of questionnaires vs the gold standard assessment of a clinical interview with the CAPS-4 or CAPS-5, nor do the authors indicate if other aspects were identified in the various studies eg measures of depression, particular post natal depression or other anxiety disorders. These aspects are particularly relevant given the authors report the administration of an enema, the birthing position, ARM without consent and SVD as linked to the CB-PTSD. 

The authors state that CB-PTSD seems to decrease over time, partly due to seeking professional help but do not seem to have provided evidence for this statement.

I found table 2 lacked clarity eg what is meant by statistically crude association and how does this differ from the statistically significant adjusted. Are the differences measured using 'p' values or Odds Ratios?

It would be helpful if a table that details the findings of the studies in this review could be included in the paper. The table should include the mean scores for the participants, the confidence intervals, and other measures used eg DASS, K10, Edinborough PND scale intrapartum traumas and risk factors (if measured).

Author Response

Reviewer #1

Comment

Response

1. Lines 18 and 19: Knowledge on the risk factors .... is researched thoroughly. Can the authors clarify this statement ie do you mean this study is researching the subject thoroughly, or are you stating the literature has identified the risk factors for CB-PTSD and that the risk factors identified relate to the likelihood of developing symptoms only in the short-term? 

We agree with the reviewer that we should have stated this more clearly in the abstract. Therefore, we changed ‘Knowledge on the risk factors of developing CB-PTS/D and its effects in the short-term is researched thoroughly’ to

‘Literature has identified the risk factors for developing CB-PTS/D within the first six months postpartum thoroughly.’

2. I note the criteria for the diagnosis of PTSD have changed with the introduction of the DSM-V. Could the authors comment as to whether this affects the prevalence of CB-PTSD?

Thank you for pointing out this issue. However, we do not have data to be able to state to what extent might the change of PTSD criteria affect the estimated prevalence of CB-PTS/D. We have addressed this issue in the Discussion section.

‘It is not clear yet whether changes in the DSM criteria introduced in DSM-V result in changes in prevalence of CB-PTS/D (44). Removal of the personal experience from the criterion A, which describes the traumatic event, might increase the number of births considered as traumatic (44,90), which could mean that more women might be diagnosed with CB-PTS/D.’

We also clarified the change of criterion A in the materials and method section (2.3 PTSD symptomatology criteria) to make the added points in the discussion more clear.

‘In DSM-IV, criterion A, which identifies the traumatic event, consists of two parts: experiencing an event that involved an actual or threatened death or serious injury and the personal response to the event with fear, helplessness or horror. The personal response to the event was removed in DSM-V (43,45). Additionally, a symptom cluster referring to negative alterations in cognition and mood (e.g. feelings of detachment, inability to experience positive emotions) was added in DSM-V (42,43). A further description of the DSM-IV and DSM-V criteria and their differences is provided by Pai et al., and in the APA factsheet (43,45).’

3. The authors indicate that PTSD symptoms or the diagnosis vary between 0.7% and 42% in the studies included in the analysis and contrast this with the results of western prevalence studies that suggest 2-9%. Is there a reason why the marked discrepancy between these figures?

The reviewer is correct in saying that the interval for PTSD diagnosis is much broader in the included studies compared to the prevalence reported in western studies generally. We addressed this issue in the Discussion section.

‘The discrepancy between the CB-PTS/D prevalence found in the studies included in this systematic review and that reported in the literature (4–7) can be explained by the fact that one study included in this review reported a very high number of women with CB-PTS/D (54). This study had however a relatively small sample size (n=102) and used a recently developed measure to screen for PTSD that was not yet used in other studies (54). Except for this study, the prevalence rate for CB-PTS/D reported in the studies included in this review is consistent with the range reported in the literature’.

In addition, the wide variability in CB-PTS/D prevalence is addressed in the Limitations.

Second, aside from selection bias and response bias in self-reporting questionnaires leading to over or underreporting of PTSD symptoms, assessing CB-PTS/D with different questionnaires in different studies resulted in varying prevalence rates of CB-PTS/D.’

4. The authors have not commented on issues related to the use of questionnaires vs the gold standard assessment of a clinical interview with the CAPS-4 or CAPS-5, nor do the authors indicate if other aspects were identified in the various studies eg measures of depression, particular post natal depression or other anxiety disorders

These aspects are particularly relevant given the authors report the administration of an enema, the birthing position, ARM without consent and SVD as linked to the CB-PTSD. 

We agree that it is important to address this issue. In our review, the majority of studies had  validated questionnaires that made sure that the results are reliable. We included the sentences in the limitations:

‘Second, aside from selection bias and response bias in self-reporting questionnaires leading to over- or under-reporting of symptoms, assessing CB-PTS/D with different questionnaires in different studies resulted in varying prevalence rates of CB-PTS/D. Also, studies using clinical interviews versus questionnaires may report higher prevalence of  CB-PTS/D (95).’ 

We agree that measures of other mental disorders are of high relevance as they are associated with PTSD. Nevertheless, our focus was on PTSD specifically, therefore we did not describe the results of the studies related to other mental disorders. Although we did not focus on other mental disorders, we took them into account when assessing the quality of the included studies: studies received 0 points for criterion 4 ‘confounding’ if they did not considered other mental disorders or previous trauma. Based on this reviewer’s comment, we added the following text discussion section:

‘Additionally, some studies adjusted the associations between mode of birth and CB-PTS/D with mental health issues prior and during pregnancy (e.g. depression), the reported adjusted associations showed inconclusive results (22,58,61,70,73).’

The reason why we reported administration of enema, birthing position, ARM without consent is that it is connected to the mode of birth.

5. The authors state that CB-PTSD seems to decrease over time, partly due to seeking professional help but do not seem to have provided evidence for this statement.

We thank the reviewer for pointing this out – we agree with the reviewer that we should rephrase the sentences in the discussion section. We made the following adaptations in the discussion section:

The majority of the cohort and longitudinal studies reported a slight decrease of CB-PTS/D over a period of time from baseline to 6 or 10 months (18,54,57,68,73). These results were published between 2015 and 2022 suggesting that, given awareness regarding CB-PTS/D in both healthcare professionals and public in the last years, women were more likely to receive professional support and help. On the other hand, one longitudinal study and a meta-analysis reported, an increase in prevalence of CB-PTS/D over time from childbirth to 6 months, or from 6 months to 12 months postpartum (58,90). It was  explained that the increase was probably due to postnatal factors, such as sleep deprivation which might postpone CB-PTS/D symptoms onset and/or prevent remission (90).’

6. I found table 2 lacked clarity eg what is meant by statistically crude association and how does this differ from the statistically significant adjusted. Are the differences measured using 'p' values or Odds Ratios?

Thank you for pointing this out, we agree that the legends accompanying the Figure 3 (we changed Table 2 into Figure 3) lacked clarity, therefore we made the following adaptations: 

‘No association calculated between mode of birth and CB-PTS/D.

No statistical significant association reported between mode of birth and CB-PTS/D.

Crude statistical significant association reported between mode of birth and CB-PTS/D.

Adjusted statistical significant association reported between mode of birth and CB-PTS/D with adjustments for confounding factors (e.g. demographics, social behaviour variables or mental health issues prior and during pregnancy).

We also adapted the paragraph in the Materials and Methods (2.6 Data extraction and management) section to enhance clarity about the reported results :

‘The following information was extracted from the included articles and reported using the Population Exposure, Comparator and Outcomes (PECO) framework (53) : author, year of publication, country, population (i.e. number of participants, ethnicity, age and parity), exposure/comparator (mode of birth), outcomes (the measure used for assessing CB-PTS/D including the interval from birth to CB-PTS/D measurement (i.e. ≥ 6 months), and the main findings. The main findings report crude and adjusted associations expressed as odds ratio with 95% confidence intervals (OR (95% CI) calculated with univariable and multivariable logistic regression analyses. Additionally, we reported correlations (calculated with Spearman Rank or Pearson correlation coefficient) and comparison of CB-PTS/D  measurement scores (calculated with Student’s t-test or ANOVA).’

7. It would be helpful if a table that details the findings of the studies in this review could be included in the paper. The table should include the

- mean scores for the participants,

- the confidence intervals, and

other measures used eg DASS, K10, Edinborough PND scale intrapartum traumas and risk factors (if measured).

In Table 1, we reported mean scores of measures of CB-PTS/D, moreover we provided, associations (with confidence intervals if given) or correlations between mode of birth and CB-PTS/D. However, given the focus of our review, we believe that reporting means of other measures (e.g. depression) would divert attention from the main topic. Moreover, not all included studies reported the scores of measures of depression and/or anxiety.

Reviewer 2 Report

Thank you for giving me the opportunity to review the manuscript.

I think it is necessary to revise the manuscript before publication.

1) In the method section, please describe whether unpublished data/material were included in this review, and of what kind of languages the authors included in the studies.

2) Please attach PRISMA 2020 checklist and fill in the page numbers.

3) To make the studies visible, please show all the 28 studies in the table, by using PECO format. Please explain the study design, the population, the exposure, the control, the outcomes, sample sizes, and the confounders by using the Table. The confounders should include a history of mental health problem, previous PTSD and trauma (sexual abuse or domestic violence), lack of respect and involvement in decision-making, unsupportive attitude of caregivers, and lack of emotional support provided to the laboring woman.

4) Because the authors reviewed many different studies, Each of the study design (cross-sectional or cohort), sample size, and pre-defined confounders  affected PTSS scores. Please explain more clearly how these factors in each study affect the PTSS scores in the discussion section.

5) In the section of Implications for future research and recommendations, the authors explain large-scale longutidunal studies were warranted, but it is still unclear how sample size is estimated, and what kind of confounders should be included  in future studies. Please clearly state how future studies should be, by using the PECO format (population, exposure, control, and outomes, and confounder).

6) The theme of this article is " the influence of Mode of Birth on Childbirth-Related Post Traumatic Stress Symptoms Beyond 6 Months Postpartum". However, this integrative review indicates that this association has not been established. To focus on investigating this association, I think it is better to delete the following sentences in the section of Implications for future research and recommendations.

"Also, future research should focus on the associations between models of perinatal are (such as continuity of midwifery care throughout pregnancy and perinatal period)  and the possible reduction of CB-PTS/D. Finally, future research could develop screening tools that identify the optimal timing of assessing CB PTS/D in the postpartum period, including providing (psychological) care. A potential tool could be to ask women two days postpartum the following question ‘Today, what are your memories of your  childbirth?’. This question was used by one study and the answers given were shown to be strongly associated with PTSD a year after childbirth . Also, health care professionals should receive training in trauma informed and humanized care to be able to provide adequate support to the laboring women." 

I think it is necessary to revise the manuscript before publication.

Reviewer 3 Report

Despite being a good study, there is a need to make changes to improve its quality.

1. In reference 39 they mention the PRISMA of 2009, there is a new version that they must follow.

2. It should be noted how many studies were found in each database and how many were chosen.

3. They write this sentence "A further description of the DSM-IV and DSM-V criteria and their differences is provided in the Appendix" but it does not appear in the text.

4. The CASP are different for the two types of studies you have chosen. In addition to the fact that the references must be different, the number of questions are different. The CASP model for qualitative studies has 10 questions. The appropriate instrument must be reviewed.

5. The cross-sectional studies, which you comment on in the results that have entered you, must have another type of quality instrument specific to them.

6. In the study by Barnett et al. minimum thresholds were defined for the criteria. In your case they have not been defined.

7. Explain all the tests that are done specifically, you cannot put "with e.g."

8. The flowchart is wrong, it does not follow the latest version. In addition, even if it is included by another means, the article must appear in identification. Change image.

9. When they say "papers using CASP-criteria" in strengths, it cannot be because quality assessment must always be done. In addition, the appropriate models have not been passed to all the articles.

10. The conclusions should be improved. They must be a response to the objectives and should not be cited in the conclusions.

Round 2

Reviewer 1 Report

Thank you for the opportunity to re-review the paper. The issues have been addressed. 

I have no further comments.

Reviewer 2 Report

Thank you for giving me the opportunity to review this revised manuscript.

I think this manuscript would be suitable for publication.